# Neutrophil-to-lymphocyte ratio may predict clinical relapse in ulcerative colitis patients with mucosal healing

**Noriyuki Kurimoto[1], Yu Nishida[2], Shuhei Hosomi [2]\*, Shigehiro Itani[2], Yumie Kobayashi[2], Rieko Nakata[2], Masaki Ominami[2], Yuji Nadatani[2], Shusei Fukunaga[2], Koji Otani[2], Fumio Tanaka[2], Yasuaki Nagami[2], Koichi Taira[2], Noriko Kamata[2], Yasuhiro Fujiwara[2]**

1 Department of Gastroenterology, Osaka City University Graduate School of Medicine, Osaka, Japan,
2 Department of Gastroenterology, Osaka Metropolitan University Graduate School of Medicine, Osaka, Japan

\* shuhosomi@gmail.com

**Data Availability Statement:** All relevant data are within the paper and its Supporting Information files.

## Abstract

Endoscopic mucosal healing (MH) is an important treatment goal for patients with ulcerative colitis (UC). The neutrophil-to-lymphocyte ratio (NLR) reflects systemic inflammation and has been reported to be a useful predictive marker for UC. This study aimed to evaluate the clinical utility of the NLR for predicting clinical relapse in UC patients with MH. We retrospectively enrolled patients with UC who underwent colonoscopy at the Osaka City University Hospital between January 2010 and December 2010, whose Mayo Endoscopic Subscore was 0 or 1. The correlation between the incidence of relapse and demographic factors, including the NLR, was analyzed. We included 129 patients in the present study. The median NLR at the time of endoscopy was 1.98, and differences in the high NLR group and the low NLR group were compared. During a median follow-up period of 46.4 months, 58 patients (45.0%) experienced relapse. The cumulative relapse-free rate was significantly higher in the low NLR group than in the high NLR group ($P$ = 0.03, log-rank test). Multivariate analysis identified high NLR as an independent prognostic factor for clinical relapse (hazard ratio, 1.74; 95% confidence interval, 1.02–2.98; $P$ = 0.04). NLR is a novel and useful predictor of clinical relapse in UC patients with MH, and it can potentially be a strong indicator to determine the appropriate treatment strategy and decision-making in clinical practice.

## Introduction

Ulcerative colitis (UC) is a chronic inflammatory bowel disease (IBD) of unknown cause that primarily affects the mucous membranes, often forming erosions and ulcers. Symptoms of ulcerative colitis include rectal bleeding and abdominal pain, and these symptoms are characterized by remission and recurrence even with sufficient treatment [1]. The European Crohn's and Colitis Organization guidelines states that achieving mucosal healing (MH) is an important goal of medical therapy [2], as MH is inversely correlated with subsequent clinical relapse, the frequency of hospitalization and surgery, and colorectal cancer incidence [3–5].

**Funding:** The author received no specific funding for this work.

**Competing interests:** The authors have declared that no competing interests exist.

Nevertheless, some patients with MH may also experience relapse. Therefore, several studies have attempted to identify the risk factors, such as histological findings and fecal calprotectin, for relapse in UC after achieving MH [6–9].

The neutrophil-to-lymphocyte ratio (NLR) is a simple index that can be calculated from the results of ordinary blood tests; it is calculated from a blood sample by dividing the absolute neutrophil count by the absolute lymphocyte count [10]. As the NLR can reflect the systemic status of inflammation or immune response, some studies have reported the association between NLR and the subsequent outcome and prognosis of cancer patients [11–13], disease activity in rheumatoid arthritis, or prognosis in septic patients [14, 15]. The usefulness of NLR has also been reported in the field of IBD; NLR is an independent prognostic factor for infliximab therapy [16] or tacrolimus therapy [17]. Akpinar et al., reported that among patients with UC, the mean values of NLR in the endoscopically active disease group were higher as compared to others, with higher values in the endoscopic remission group than in the control group [18]. However, no studies have evaluated the value of NLR in predicting clinical relapse in UC patients with MH. Therefore, our study aimed to assess the clinical utility of the NLR for predicting clinical relapse in patients with UC who have achieved MH.

## Materials and methods

### Patients

We retrospectively included patients with UC who underwent colonoscopy at the Osaka City University Hospital between January 2010 and December 2010, whose Mayo Endoscopic Subscore (MES) was 0 or 1. We excluded patients who did not have laboratory data on differential white blood cell (WBC) count at endoscopy, patients who changed treatment within 6 months prior to endoscopy, patients who received treatment other than mesalazine (e.g., corticosteroids, immunomodulator, anti-tumor necrosis factor (TNF)-α antibody, and calcineurin inhibitor), and patients without clinical remission.

### Evaluation

All patients were followed-up with a physical examination and a blood test. The differential WBC count was analyzed using an XE-5000 hematology analyzer (Sysmex, Kobe, Japan), as per the manufacturer's protocol. Patients were followed-up from the time of colonoscopy to the onset of clinical relapse, until they were lost to follow-up, or until the end of June 2021. The NLR was calculated from a blood sample by dividing the absolute neutrophil count by the absolute lymphocyte count. In patients who attenuated treatment based on clinical symptoms or endoscopic findings, we defined the tracking period from the time of colonoscopy to the onset of attenuating treatment.

### Definitions

Clinical relapse was defined as the exacerbation of gastrointestinal symptoms requiring secondary alternative therapies such as surgery, administration of corticosteroids, or biologics. MH was defined as an MES of 0 or 1 [19].

### Study endpoints

The primary outcome of this study was clinical relapse. Predictors of clinical relapse, including various demographic and clinical variables, such as NLR, were analyzed.

## Statistical analysis

Continuous variables are presented as medians and interquartile ranges (IQR). The differences in clinical characteristics were compared using either the Chi-squared test or Fisher's exact test for categorical variables and the Mann-Whitney U-test for continuous variables. The median value was constructed to define a cutoff level for each parameter. The non-cumulative incidence of clinical relapse was illustrated using a Kaplan-Meier plot. Differences in the survival curves were assessed using the log-rank test. A multivariate analysis was performed using a Cox regression model. Data were presented as hazard ratios (HRs) with 95% confidence intervals (CIs). Multivariate Cox regression analyses were performed to identify factors associated with clinical relapse; those factors speculated to be risk factors for clinical relapse were then evaluated in the multivariate analysis.

A P-value of $< 0.05$ was considered statistically significant. All statistical analyses were performed with EZR (Saitama Medical Center, Jichi Medical University), a graphical user interface for R (The R Foundation for Statistical Computing, version 4.1.1). More precisely, it is a modified version of R commander (version 2.7–1) that includes statistical functions frequently used in biostatistics.

## Ethical considerations

This study was approved by the Osaka City University Hospital Certified Review Board (no. 2020–008), which waived the requirement for written informed consent because the analysis used anonymized clinical data that were retrospectively obtained after each patient agreed to receive the treatment. Nevertheless, all patients were notified of the content and information of this study and were given the opportunity to refuse participation. None of the patients refused participation. This study followed the Ethical Guidelines for Medical and Health Research Involving Human Subjects established by the Ministry of Education, Culture, Sports, Science and Technology, and the Ministry of Health, Labor and Welfare in Japan.

## Results

### Study subjects

Overall, we included 253 patients with UC who underwent colonoscopy, with an MES of 0 or 1 during the study period. Among them, 43 patients without data on differential WBC count, 18 patients who changed treatment within 6 months prior to endoscopy, eight patients who received immunomodulators, 38 patients who received corticosteroids, eight patients who were in a clinical trial for the treatment of UC, two patients who had outpatient care at other hospitals after endoscopy, and one patient who underwent cytapheresis (CAP) treatment were excluded. Additionally, eight patients without clinical remission on their colonoscopy were also excluded. Finally, 129 patients were retrospectively reviewed.

The median follow-up period was 46.4 months (IQR: 9.0–95.7 months). Fifty-eight patients (45.0%) experienced clinical relapse during the observation period. The median duration of remission for the 79 patients who remained in remission was 70.9 months (IQR: 12.3–125.8 months). The median duration of remission for the 58 patients who relapsed was 29.0 months (IQR: 6.5–65.5 months). The cumulative relapse-free rate was 82.8%, 76.2%, and 46.4% at 12, 24, and 96 months, respectively (Fig 1). The demographic characteristics of the patients are summarized in Table 1.

### Comparison between the high NLR and low NLR groups

As the median NLR was 1.98 (IQR: 1.39–2.59), high NLR was defined as an NLR value $\geq 1.98$. Sixty-five (50.4%) patients had a high NLR. The comparison between the low NLR group and

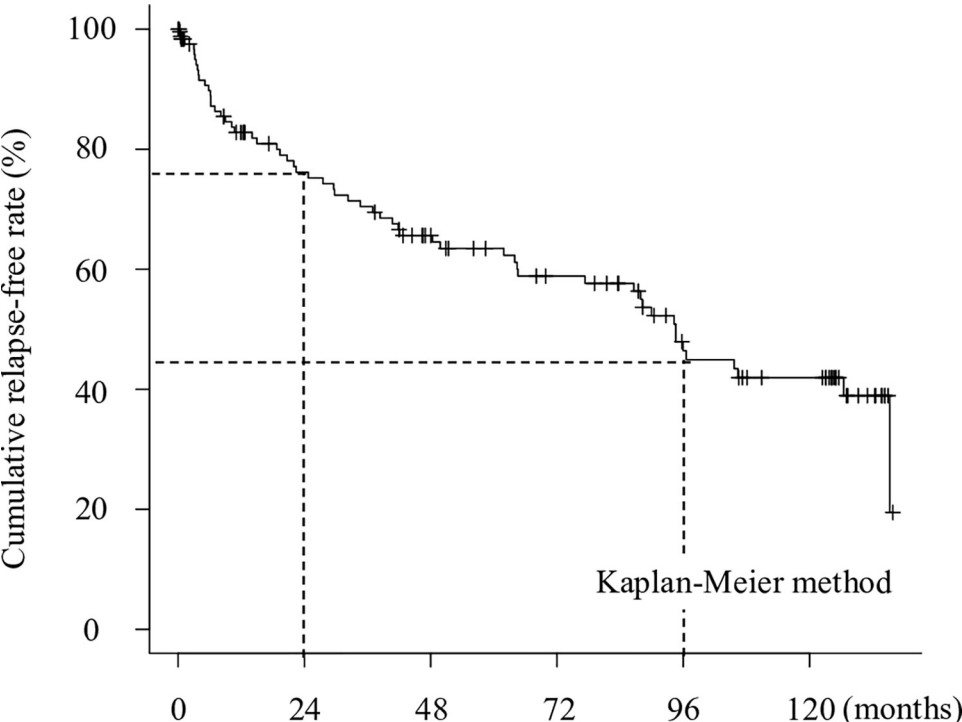

**Fig 1. Cumulative incidence of clinical relapse.** The cumulative relapse-free rate was 76.2% and 46.4% at 24 and 96 months, respectively.

**Table 1. Baseline characteristics of the study population.**

|  | all patients |
|---|---|
| Number of patients | 129 |
| Gender: male / female | 67 / 62 |
| Age at diagnosis (years) | 31.9 (25.7–41.7) |
| Disease duration (years) | 12.3 (6.8–18.8) |
| UC location: Left-sided colitis / Pancolitis / Proctitis | 45 / 57 / 27 |
| MES score: 0 / 1 | 91 / 38 |
| 5-ASA or SASP therapy, n (%) | 123 (95.3%) |
| Hemoglobin (g/dL) | 13.9 (12.9–15.0) |
| Albumin (g/dL) | 4.20 (4.10–4.40) |
| CRP (mg/dL) | 0.04 (0.02–0.09) |
| WBC (/μL) | 5700 (4700–6800) |
| Neutrophil (/μL) | 3234 (2636–4167) |
| Lymphocyte (/μL) | 1694 (1470–2111) |
| NLR | 1.98 (1.39–2.59) |
| Platelet ($10^4$/μL) | 21.6 (18.8–26.0) |
| Clinical relapse, n (%) | 58 (45.0%) |

data are expressed as median (interquartile range) for continuous variables and as numbers (percentage) for categorical variables

UC: ulcerative colitis; MES: Mayo endoscopic subscore; 5-ASA: 5-aminosalicylic acid; SASP: salazosulfapyridine; CRP: C-reactive protein; WBC: white blood cell; NLR: neutrophil-to-lymphocyte ratio.

**Table 2. Comparison between low NLR and high NLR.**

| | low NLR | high NLR | *P*-value |
|---|---|---|---|
| Number of patients | 64 | 65 | |
| Gender: male/female | 32 / 32 | 35 / 30 | 0.73 |
| Age at diagnosis (years) | 33.0 (26.7–41.8) | 30.1 (24.9–40.2) | 0.79 |
| Disease duration (years) | 13.0 (6.7–19.0) | 11.2 (6.9–18.8) | 0.77 |
| UC location: Left-sided colitis / Pancolitis / Proctitis | 21 / 26 / 17 | 24 / 31 / 10 | 0.31 |
| MES score: 0 / 1 | 48 / 16 | 43 / 22 | 0.34 |
| 5-ASA or SASP therapy, n (%) | 62 (96.9%) | 61 (93.8%) | 0.68 |
| Hemoglobin (g/dL) | 14.0 (12.8–14.9) | 13.7 (13.1–15.1) | 0.8 |
| Albumin (g/dL) | 4.30 (4.10–4.40) | 4.20 (4.00–4.30) | 0.04 |
| CRP (mg/dL) | 0.03 (0.01–0.06) | 0.04 (0.02–0.11) | 0.07 |
| WBC (/μL) | 5250 (4500–5800) | 6300 (5300–7400) | <0.001 |
| Neutrophil (/μL) | 2686 (2312–3109) | 4092 (3473–4884) | <0.001 |
| Lymphocyte (/μL) | 2006 (1644–2316) | 1578 (1130–1794) | <0.001 |
| NLR | 1.39 (1.15–1.63) | 2.59 (2.23–3.29) | <0.001 |
| Platelet ($10^4$/μL) | 21.6 (19.4–25.8) | 22.2 (18.3–26.7) | 0.79 |
| Clinical relapse, n (%) | 24 (37.5%) | 34 (52.3%) | 0.09 |

data are expressed as median (interquartile range) for continuous variables and as numbers (percentage) for categorical variables

UC: ulcerative colitis; MES: Mayo endoscopic subscore; 5-ASA: 5-aminosalicylic acid; SASP: salazosulfapyridine; CRP: C-reactive protein; WBC: white blood cell; NLR: neutrophil-to-lymphocyte ratio.

the high NLR group is shown in Table 2. Compared with patients with low NLR, those with high NLR had significantly higher WBC counts ($P < 0.001$) and neutrophil counts ($P < 0.001$), and significantly lower lymphocyte counts ($P < 0.001$) and albumin levels ($P = 0.03$). Fig 2 shows a comparison of the cumulative relapse-free rate between the low NLR and high NLR groups. The cumulative relapse-free rate was significantly higher in the low NLR group than in the high NLR group ($P = 0.03$, log-rank test). The cumulative relapse-free rate in the high NLR group was 67.8% and 35.8% at 24 and 96 months, respectively, and that in the low NLR group was 84.4% and 53.4% at 24 and 96 months, respectively.

## Risk factors for clinical relapse

After achieving MH, demographic variables were evaluated as prognostic factors for clinical relapse by univariate Cox regression analysis. High NLR was significantly associated with clinical relapse (unadjusted HR, 1.78; 95% CI, 1.05–3.01; $P = 0.03$). Other clinical variables such as sex, age of onset, disease duration, site of disease, MES, and platelet count did not show a statistically significant association with clinical relapse (Table 3).

To elucidate the influence of speculated risk factors, multivariate Cox regression analysis was performed to identify factors associated with clinical relapse. Variables included in the multivariate analysis were sex, MES, and platelet count. Multivariate Cox regression analyses identified high NLR as an independent prognostic factor for clinical relapse (adjusted HR, 1.74; 95% CI, 1.02–2.98; $P = 0.04$) (Table 4).

## Discussion

In this study, we showed that in UC patients with MH, the NLR was a predictor of subsequent clinical relapse. NLR is a minimally invasive marker measured by a routine blood test. There

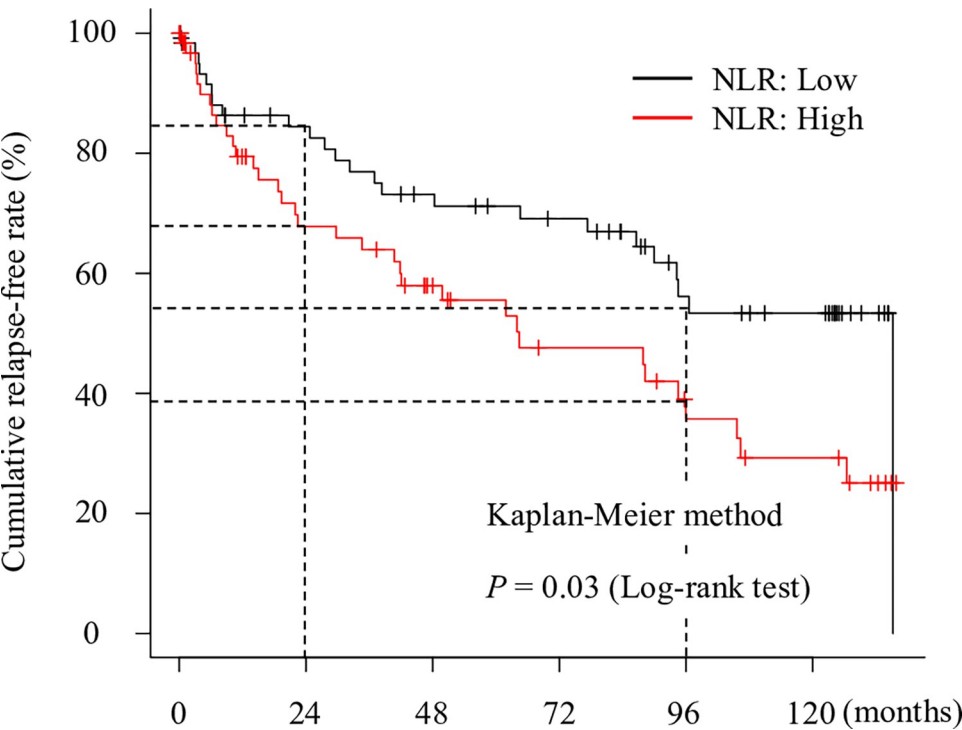

**Fig 2. Comparison of cumulative incidence of clinical relapse between the high and low neutrophil-to-lymphocyte ratio (NLR) groups.** The cumulative relapse-free rate in the high NLR group was 67.8% and 35.8%, and the cumulative relapse-free rate in the low NLR group was 84.4% and 53.4% at 24 and 96 months, respectively. The cumulative relapse-free rate was significantly higher in the low NLR group than in the high NLR group ($P$ = 0.03, log-rank test).

have been reports on the association of NLR with therapeutic agents for UC, including reports that NLR may be a biomarker for predicting the outcome of systemic corticosteroid therapy [20] and that NLR is an early predictor of therapeutic response to TNF-α antibody therapy [21]. However, to the best of our knowledge, the prediction of clinical relapse in UC patients with MH using NLR has not been investigated previously. In this study, we described the significance of NLR in patients who achieve MH with no medication or with 5-ASA/SASP preparations alone. Therefore, the present results can be used as criteria for increasing or decreasing the dose of the 5-ASA/SASP formulation, but not as criteria for switching to other UC therapies. We believe that if remission is maintained and the NLR is low, reducing the dose of the therapeutic agent can be considered. Conversely, if remission is maintained but the NLR is high, it can be used as a criterion for increasing the dose of the therapeutic agent.

The underlying mechanism that the NLR was a predictor of clinilcal relapse in UC patients with MH remains poorly understood. Neutrophils induce cytotoxicity and inflammation in UC. A high NLR value may be due to high activity in UC and lowered mucosal barrier function, which may lead to the migration of neutrophils through the gut microbiota [22, 23]. Regarding lymphocytes, they are also involved in the immune function of the host [24]. Furthermore, malnutrition is associated with lymphocyte depletion [25], and malnutrition has been reported to be a predictor for disease flare [26]. It is possible that the depletion of lymphocytes induces clinical relapse due to malnutrition. In the present study, the high NLR group had low albumin level and albumin level was not a predictor of relapse in the univariate analysis using the cox proportional hazards model, and there were no cases of hypoalbuminemia in the blood tests.

**Table 3. Cox regression analysis of risk for clinical relapse.**

| | Unadjusted HR (95% CI) | *P*-value |
|---|---|---|
| Gender | | |
| Male | 1 | |
| Female | 1.07 (0.64–1.80) | 0.8 |
| Age at diagnoisis (continuous) | 1.00 (0.98–1.02) | 0.89 |
| Disease duration (continuous) | 0.97 (0.95–1.00) | 0.08 |
| MES score | | |
| 0 | 1 | |
| 1 | 1.33 (0.78–2.28) | 0.3 |
| UC location | | |
| left-sided colitis | 1 | |
| pan-colitis | 0.99 (0.57–1.74) | 0.98 |
| proctitis | 1.30 (0.70–2.43) | 0.41 |
| 5-ASA or SASP therapy | | |
| No | 1 | |
| Yes | 0-Inf | 0.996 |
| Hemoglobin (continuous) | 0.89 (0.74–1.07) | 0.23 |
| Albumin (continuous) | 0.55 (0.19–1.60) | 0.27 |
| CRP (continuous) | 1.45 (0.85–1.60) | 0.27 |
| NLR | | |
| low | 1 | |
| high | 1.78 (1.05–3.01) | 0.03 |
| Neutrophil (continuous, per 1000/μL) | 1.00 (0.98–1.00) | 0.73 |
| Lymphocyte (continuous, per 1000/μL) | 1.00 (0.99–1.00) | 0.25 |
| Platelet (continuous, per $10^4$/μL) | 0.97 (0.95–1.00) | 0.9 |

data are expressed as median (interquartile range) for continuous variables and as numbers (percentage) for categorical variables

HR: hazard ratio; CI: confidential interval; UC: ulcerative colitis; MES: Mayo endoscopic subscore; 5-ASA: 5-aminosalicylic acid; SASP: salazosulfapyridine; CRP: C-reactive protein; NLR: neutrophil-to-lymphocyte ratio.

**Table 4. Cox proportional hazards regression of risk for clinical relapse.**

| | Adjusted HR (95% CI) | *P*-value |
|---|---|---|
| Gender | | |
| Male | 1 | |
| Female | 1.17 (0.67–2.04) | 0.59 |
| MES score | | |
| 0 | 1 | |
| 1 | 1.25 (0.71–2.20) | 0.44 |
| NLR | | |
| low | 1 | |
| high | 1.74 (1.02–2.98) | 0.04 |
| Platelet (continuous, per $10^4$/μL) | 0.99 (0.95–1.03) | 0.55 |

HR: hazard ratio; CI: confidential interval; MES: Mayo endoscopic subscore; NLR: neutrophil-to-lymphocyte ratio.

We identified a cutoff value of NLR of 1.98 to predict relapse after achieving MH. Lorenzo et al., evaluated NLR at baseline in patients with UC who were treated with infliximab and showed that a value of 2.06 was predictive of MH after 54 weeks of infliximab treatment [21]. This cutoff value was relatively similar to the one obtained in the present study. It is difficult to compare the cutoff value of these studies owing to the differences in the study design. Nonetheless, the cutoff values of this study may be appropriate for the examination of UC for MH.

Because the sample size of this study was relatively small, the general formula for calculating the total sample size was used to calculate the required sample size [27]. Sample size calculation indicated that a total of 118 patients were required to detect a significant association between NLR and clinical relapse with the following assumptions: an α level of 0.05 and a β level of 0.20; half of the patients were allocated to the high NLR group, the incidences of non-clinical relapse in patients with high and low NLR group were 35.8% and 53.4%, and the follow-up period was 8 years. Therefore, the sample size in this study was sufficient to examine the association between NLR and the development of clinical relapse.

Nakarai et al., reported that platelet count was a predictor of clinical relapse in UC patients with MH [28]; however, in the present study, platelet count was not found to be a risk factor for clinical relapse.

The use of 5-aminosalicylic acid (5-ASA) or salazosulfapyridine (SASP) can induce neutropenia [29, 30]. Neutropenia tends to occur at a SASP dose of 3.0–4.0 g/day and within the first 2 months of administration [31]. In the present study, 74 patients were taking 5-ASA and 44 patients were taking SASP. However, there was no significant difference in the use of 5-ASA or SASP between the high NLR and low NLR groups, suggesting that the influence of 5-ASA or SASP medication on NLR might be negligible.

In regard to endoscopic activities, it is controversial whether an MES of 1 is a risk factor for relapse. Some studies state that an MES of 1 is a risk factor for relapse as compared to an MES of 0 [32, 33], whereas other studies have shown contradictory results [4, 34]. In the present study, an MES of 1 was not identified as a risk factor for clinical relapse and there are several possible reasons. First, the sample size of the present study was relatively small. Second, the present study only included patients with clinical remission, unlike most previous reports. Furthermore, this study excluded patients using corticosteroids, immunomodulators, or molecular-targeted therapies, which could influence WBCs.

In this study, the remission maintenance rates in UC patients with MH were 82.8% at 12 months. Previous studies have reported a remission maintenance rate of 73–86% at 12 months in UC patients with MH [28, 33, 35, 36]. Therefore, the remission maintenance rate at 12 months obtained in this study is comparable to previous studies.

Our study has some limitations. This was a single-center, retrospective study with a relatively small cohort that is susceptible to bias in data selection and analysis. NLR is affected by many factors, including malignancy, coronary artery disease, and infection [37–39]. We checked the medical records for underlying diseases other than UC and excluded cases with complications such as Sjögren's syndrome. In addition, there were no cases of infectious disease immediately before enrollment based on our review of the medical records. However, we cannot completely rule out the possibility that patients were treated for colds or other infectious diseases at the hospital they were visiting for other underlying diseases. It is also possible that the patients' UC symptoms flared up during the observation period due to some infection or worsening of other comorbidities. Furthermore, we did not consider the influence of medications that could change the neutrophil count, such as histamine type-2 receptor antagonists or non-steroidal anti-inflammatory drugs. In addition, we could not evaluate potentially predictive factors, such as histological findings, fecal calprotectin [6–9], prostaglandin E-major urinary metabolite [40], or leucine-rich alpha-2 glycoprotein [34], since this was a

retrospective study. As the number of elderly UC patients is increasing [41], the number of patients with comorbidities, immune dysfunction, polypharmacy, and other factors affecting NLR values is also increasing. Prospective multicenter studies seem necessary to confirm the usefulness of NLR in the real world. Finally, this study did not include patients receiving treatment other than 5-ASA or SASP. We enrolled patients who underwent colonoscopy in 2010. Molecular-targeted therapies are now a major therapeutic strategy for the treatment of UC; however, most of them, including golimumab, ustekinumab, vedolizumab, and tofacitinib, have emerged after 2010. Therefore, a multicenter prospective study with a larger sample size may help verify our results and confirm whether NLR is a predictor of clinical relapse after MH.

In this study, we investigated the utility of NLR in predicting clinical relapse in patients with UC who have achieved MH. Our results suggest that a high NLR is associated with an increased risk of clinical relapse in UC patients with MH, and patients with a high NLR should be followed-up carefully.

## Supporting information

**S1 Data.**
(XLSX)

## Author Contributions

**Conceptualization:** Shusei Fukunaga.

**Supervision:** Shuhei Hosomi.

**Writing – original draft:** Noriyuki Kurimoto.

**Writing – review & editing:** Yu Nishida, Shigehiro Itani, Yumie Kobayashi, Rieko Nakata, Masaki Ominami, Yuji Nadatani, Shusei Fukunaga, Koji Otani, Fumio Tanaka, Yasuaki Nagami, Koichi Taira, Noriko Kamata, Yasuhiro Fujiwara.

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
