## [Decision Letter · Decision Letter 0]

14 Sep 2022

PONE-D-22-16251Neutrophil-to-lymphocyte ratio may predict clinical relapse in ulcerative colitis patients with mucosal healingPLOS ONE

Dear Dr. Hosomi

Thank you for submitting your manuscript to PLOS ONE. After careful consideration, we feel that it has merit but does not fully meet PLOS ONE’s publication criteria as it currently stands. Therefore, we invite you to submit a revised version of the manuscript that addresses the points raised during the review process.

We have now received comments from the referee of your manuscript, we invite you to submit a revised version of the manuscript. Please consider and address each of the comments raised by the reviewer.  

We look forward to receiving your revised manuscript.

Kind regards,

Senthilnathan Palaniyandi, Ph.D

Academic Editor

PLOS ONE

Journal Requirements:

Reviewers' comments:

Reviewer's Responses to Questions

**Comments to the Author**

1. Is the manuscript technically sound, and do the data support the conclusions?

Reviewer #1: Yes

Reviewer #2: Partly

2. Has the statistical analysis been performed appropriately and rigorously? 

Reviewer #1: I Don't Know

Reviewer #2: No

3. Have the authors made all data underlying the findings in their manuscript fully available?

Reviewer #1: Yes

Reviewer #2: Yes

4. Is the manuscript presented in an intelligible fashion and written in standard English?

Reviewer #1: Yes

Reviewer #2: Yes

5. Review Comments to the Author

Reviewer #1: The manuscript is well written, however still a novel score but available to every patient, but we don't really know if it will be used in the clinical context. In addition, this is a single center study and additional studies are needed.

Reviewer #2: Dear Authors these are my comments;

1.The authors posit that NLR can be used to determine treatment option. Can you please further elaborate how NLR can provide insight into the selection of agents for UC

2. Sample Size calculation- The authors have not depicted their sample size calculation to ensure that the number of subjects (n) is enough for a statistically well powered study.

3.Patient subsection of materials and methods section: It will be interesting to know how many of the 129 subjects have been maintaining remission and for how long or was remission only received in the past few months. This can provide insight if NLR can also predict durable remission.

4.Can you please clarify if selected patients had been screened for any other comorbidities such as an ongoing infection that may affect the NLR. This is a major limitation to the use of NLR as it can easily be affected by infections or sometimes even states of inflammation. Practically, IBD patients will be on corticosteroids as well and this again may hamper the use of NLR.

6. PLOS authors have the option to publish the peer review history of their article (what does this mean?). If published, this will include your full peer review and any attached files.

Reviewer #1: No

Reviewer #2: No

---

## [Author Response · Author response to Decision Letter 0]

2 Nov 2022

Response to the Reviewer #1 comments:

We appreciate your constructive comments and valuable suggestions as they helped us to revise and improve this manuscript. As indicated in the responses that follow, we have considered these comments and suggestions in the revised version of our manuscript. All changes in the manuscript in response to the critiques are indicated with yellow highlights.

Reviewer #1: The manuscript is well written, however still a novel score but available to every patient, but we don't really know if it will be used in the clinical context. In addition, this is a single center study and additional studies are needed.

Thank you for your valuable remarks. We are in full agreement that this study is a retrospective and whether it would be would be used in clinical context. Actually, the number of elderly UC patients is increasing, and it is likely that more cases will be affected by underlying diseases and medications that affect NLR values. In the future, it is desirable to conduct prospective multicenter studies to confirm the usefulness of NLR in the real world, however; this would be beyond the scope of the present study. The following text was added to Discussion: "As the number of elderly UC patients is increasing [41], the number of patients with comorbidities, immune dysfunction, polypharmacy, and other factors affecting NLR values is also increasing. Prospective multicenter studies seem necessary to confirm the usefulness of NLR in the real world." (line 228-231)

"Therefore, a multicenter prospective study with a larger sample size may help verify our results and confirm whether NLR is a predictor of clinical relapse after MH." (line 235-237)

Response to the Reviewer #2 comments:

We appreciate your constructive comments and valuable suggestions that have helped improve this manuscript. As indicated in the responses that follow, we have considered these comments and suggestions, which are reflected in our revised manuscript. All changes in the manuscript made in response to the comments are highlighted in yellow.

1.The authors posit that NLR can be used to determine treatment option. Can you please further elaborate how NLR can provide insight into the selection of agents for UC

Thank you for your valuable suggestions. There have been several reports on the association between therapeutic agents for ulcerative colitis and NLR. To the best of our knowledge, there are reports that NLR can be a biomarker for predicting the therapeutic outcome of systemic corticosteroid therapy (univariate analysis only) [Endo K, et al. Inflamm Intest Dis. 2021 Nov 16;6(4):218-24] and that NLR can be an early predictor of therapeutic response to anti-TNF therapy [Bertani L, et al. Inflamm Bowel Dis. 2020 Sep 18;26(10):1579-87]. This study describes the significance of NLR in patients who achieved mucosal healing with no medication or with 5-ASA/SASP formulations alone. Therefore, the results of this study can be used as criteria for increasing or decreasing the dose of 5-ASA/SASP formulations, but not for changing to other UC treatments. Treatment attenuation might be considered for patients with mucosal healing with low NLR. We paraphrased the manuscript as follows: "There have been reports on the association of NLR with therapeutic agents for UC, including reports that NLR may be a biomarker for predicting the outcome of systemic corticosteroid therapy [20] and that NLR is an early predictor of therapeutic response to TNF-α antibody therapy [21]. However, to the best of our knowledge, the prediction of clinical relapse in UC patients with MH using NLR has not been investigated previously. In this study, we described the significance of NLR in patients who achieve MH with no medication or with 5-ASA/SASP preparations alone. Therefore, the present results can be used as criteria for increasing or decreasing the dose of the 5-ASA/SASP formulation, but not as criteria for switching to other UC therapies. We believe that if remission is maintained and the NLR is low, reducing the dose of the therapeutic agent can be considered. Conversely, if remission is maintained but the NLR is high, it can be used as a criterion for increasing the dose of the therapeutic agent." (line 161-172)

2. Sample Size calculation- The authors have not depicted their sample size calculation to ensure that the number of subjects (n) is enough for a statistically well powered study.

We are in full agreement that we should have mentioned whether the sample size is sufficient because the sample size in this study was rather small. We calculated the required sample size. Sample size calculation as a post-hoc analysis indicated that a total of 118 patients were required to detect a significant association between low NLR and the development of non-clinical relapse with the following assumption: an α level of 0.05, a β level of 0.20, half of the patients were allocated to the high NLR group, the incidences of non-clinical relapse in patients with high and low NLR group were 35.8% and 53.4%, and the follow-up period was 8 years. Therefore, the sample size in this study was satisfied to examine the association between NLR and the development of clinical relapse. We have added the following sentences in the Discussion: "Because the sample size of this study was relatively small, the general formula for calculating the total sample size was used to calculate the required sample size [27]. Sample size calculation indicated that a total of 118 patients were required to detect a significant association between NLR and clinical relapse with the following assumptions: an α level of 0.05 and a β level of 0.20; half of the patients were allocated to the high NLR group, the incidences of non-clinical relapse in patients with high and low NLR group were 35.8% and 53.4%, and the follow-up period was 8 years. Therefore, the sample size in this study was sufficient to examine the association between NLR and the development of clinical relapse." (lines 189-196)

3.Patient subsection of materials and methods section: It will be interesting to know how many of the 129 subjects have been maintaining remission and for how long or was remission only received in the past few months. This can provide insight if NLR can also predict durable remission.

Thank you very much for your valuable remarks. We fully agree with you that you should have mentioned the details of the duration of remission in patients who have achieved mucosal healing. As noted in the text, 71 of 129 patients remained in remission. The median duration of remission during the observation period was 70.87 months (IQR: 12.32-125.75 months). The 58 patients who relapsed had a median duration of remission of 28.97 months (IQR: 6.5-65.49 months). The study excluded patients within 6 months of treatment intensification when enrolling them. Therefore, all patients were checked for symptoms and treatment up to 6 months prior to the date of enrollment to assure that they were in remission. However, maintenance of remission prior to 6 months before the date of enrollment could not be confirmed in this study. The following text was added to the Result: "The median duration of remission for the 79 patients who remained in remission was 70.9 months (IQR: 12.3–125.8 months). The median duration of remission for the 58 patients who relapsed was 29.0 months (IQR: 6.5–65.5 months)." (line 116-118)

4.Can you please clarify if selected patients had been screened for any other comorbidities such as an ongoing infection that may affect the NLR. This is a major limitation to the use of NLR as it can easily be affected by infections or sometimes even states of inflammation. Practically, IBD patients will be on corticosteroids as well and this again may hamper the use of NLR.

Thank you very much for your valuable remarks. We have excluded cases that are on oral medications that may affect NLR values, such as corticosteroids or IM, as stated in the paper. We checked the medical records for underlying diseases other than UC and excluded cases with complications such as Sjögren's syndrome, for example. In addition, there were no cases that had been diagnosed with any infectious diseases shortly before enrollment, as far as we could ascertain from the medical records. However, we cannot completely rule out the possibility that the patient had been treated for other diseases, such as the common cold or some other infectious disease or taking drugs prescribed in other hospitals. It is also possible that the patient's UC symptoms flared up during the observation period due to some infection or worsening of other comorbidities, and the observation period ended. The following text was added to Discussion: "We checked the medical records for underlying diseases other than UC and excluded cases with complications such as Sjögren's syndrome. In addition, there were no cases of infectious disease immediately before enrollment based on our review of the medical records. However, we cannot completely rule out the possibility that patients were treated for colds or other infectious diseases at the hospital they were visiting for other underlying diseases. It is also possible that the patients' UC symptoms flared up during the observation period due to some infection or worsening of other comorbidities." (line 218-224)

---

## [Decision Letter · Decision Letter 1]

27 Dec 2022

Neutrophil-to-lymphocyte ratio may predict clinical relapse in ulcerative colitis patients with mucosal healing

PONE-D-22-16251R1

Dear Dr. Hosomi,

We’re pleased to inform you that your manuscript has been judged scientifically suitable for publication and will be formally accepted for publication once it meets all outstanding technical requirements.

Kind regards,

Senthilnathan Palaniyandi, Ph.D

Academic Editor

PLOS ONE

Additional Editor Comments (optional):

Reviewers' comments:

Reviewer's Responses to Questions

**Comments to the Author**

1. If the authors have adequately addressed your comments raised in a previous round of review and you feel that this manuscript is now acceptable for publication, you may indicate that here to bypass the “Comments to the Author” section, enter your conflict of interest statement in the “Confidential to Editor” section, and submit your "Accept" recommendation.

Reviewer #1: All comments have been addressed

2. Is the manuscript technically sound, and do the data support the conclusions?

Reviewer #1: Yes

3. Has the statistical analysis been performed appropriately and rigorously? 

Reviewer #1: I Don't Know

4. Have the authors made all data underlying the findings in their manuscript fully available?

Reviewer #1: Yes

5. Is the manuscript presented in an intelligible fashion and written in standard English?

Reviewer #1: Yes

6. Review Comments to the Author

Reviewer #1: The manuscript was improved after the revision. The points raised were addressed. In the future additional studies are needed due to the limitations of the study.

Now the manuscript is well written, and all section are well described.

7. PLOS authors have the option to publish the peer review history of their article (what does this mean?). If published, this will include your full peer review and any attached files.

Reviewer #1: No

---

## [Editor Report · Acceptance letter]

2 Jan 2023

PONE-D-22-16251R1 

Neutrophil-to-lymphocyte ratio may predict clinical relapse in ulcerative colitis patients with mucosal healing 

Dear Dr. Hosomi:

I'm pleased to inform you that your manuscript has been deemed suitable for publication in PLOS ONE. Congratulations! Your manuscript is now with our production department. 

Kind regards, 

on behalf of

Dr. Senthilnathan Palaniyandi 

Academic Editor

PLOS ONE